# Highly Sensitive Measurement of Horseradish Peroxidase Using Surface-Enhanced Raman Scattering of 2,3-Diaminophenazine

**DOI:** 10.3390/molecules29040793

**Published:** 2024-02-08

**Authors:** Evgeniy G. Evtushenko, Elizaveta S. Gavrilina, Alexandra D. Vasilyeva, Lyubov V. Yurina, Ilya N. Kurochkin

**Affiliations:** 1N.M. Emanuel Institute of Biochemical Physics RAS, Kosygina Str. 4, 119334 Moscow, Russiaalexandra.d.vasilyeva@gmail.com (A.D.V.); ikur@genebee.msu.ru (I.N.K.); 2Faculty of Chemistry, Lomonosov Moscow State University, Leninskie Gory 1/3, 119991 Moscow, Russia

**Keywords:** surface-enhanced Raman scattering, SERS, horseradish peroxidase, HRP, ELISA, 2,3-diaminophenazine, DAP, *o*-phenylenediamine, 1,2-diaminobenzene, oPD

## Abstract

The development of various enzyme-linked immunosorbent assays (ELISAs) coupled with surface-enhanced Raman scattering (SERS) detection is a growing area in analytical chemistry due to their potentially high sensitivity. A SERS-based ELISA with horseradish peroxidase (HRP) as an enzymatic label, an *o*-phenylenediamine (oPD) substrate, and a 2,3-diaminophenazine (DAP) enzymatic product was one of the first examples of such a system. However, the full capabilities of this long-known approach have yet to be revealed. The current study addresses a previously unrecognized problem of SERS detection stage performance. Using silver nanoparticles and model mixtures of oPD and DAP, the effects of the pH, the concentration of the aggregating agent, and the particle surface chloride stabilizer were extensively evaluated. At the optimal mildly acidic pH of 3, a 0.93 to 1 M citrate buffer, and AgNPs stabilized with 20 mM chloride, a two orders of magnitude advantage in the limits of detection (LODs) for SERS compared to colorimetry was demonstrated for both DAP and HRP. The resulting LOD for HRP of 0.067 pmol/L (1.3 amol per assay) underscores that the developed approach is a highly sensitive technique. We suppose that this improved detection system could become a useful tool for the development of SERS-based ELISA protocols.

## 1. Introduction

An enzyme-linked immunosorbent assay (ELISA) with colorimetric detection is a universal analytical platform used for the quantitative measurement of various antigens. Although highly successful and widely used, in some cases, it suffers from a limited sensitivity, as more and more antigens emerge (e.g., clinically relevant ones) that have to be detected at very low concentrations. One of the natural solutions for such instances is the replacement of colorimetry with more sensitive detection techniques such as fluorescence [1,2] or enhanced chemiluminescence [3]. Surface-enhanced Raman scattering (SERS) is also a promising candidate that meets the requirement of high sensitivity. Indeed, a SERS-based ELISA is a growing area in analytical chemistry that is gradually accumulating a range of available approaches as well as a number of successful applications [4,5,6,7,8,9,10,11,12,13,14,15,16,17,18,19,20] (see also reviews [21,22]).

Horseradish peroxidase (HRP) is the most common enzyme label used in an ELISA. It is readily available both as a pure enzyme and conjugated with antibodies. Therefore, the HRP label is the first choice for the development of a more sensitive detection technique. A direct approach for the SERS-based HRP measurement relies on the difference in spectra for enzymatic substrate and product. For highly sensitive applications, the product should have a much more intense SERS spectrum compared to the substrate, as the latter is typically present in excess. In other words, the product should have both a high Raman scattering cross-section and a high affinity for the metal surface used for enhancement. Several substrates have been tested up to date: *o*-phenylenediamine (oPD) [14,15,23,24,25,26,27,28], 2,2′-azino-bis(3-ethylbenzothiazoline-6-sulfonic acid) (ABTS) [16], 3,3′,5,5′-tetramethylbenzidine (TMB) [17,18,19], and leuco dyes [11]. Among these, oPD appears to be the most promising. Firstly, it results in a well-defined and sufficiently soluble product: 2,3-diaminophenazine (DAP) [29,30]. Secondly, this reaction is not a simple oxidation, but rather, an oxidative dimerization, leading to substantial differences in the chemical structures, and hence, in the vibrational spectra of the substrate and the product. It has also been previously shown that all forms of DAP (neutral DAP, protonated DAPH^+^, and doubly-protonated DAPH_2_^2+^) exhibit intense SERS spectra with silver nanoparticles [15,27,31,32]. Finally, both oPD and DAP are commercially available, which is not relevant for the assay but greatly simplifies the research, as model solutions of both substances and their mixtures may be easily prepared and studied.

In general, the SERS-based measurements of HRP consist of two steps: enzymatic reaction and SERS detection of the resultant product. In order to achieve a high sensitivity, the conditions for both steps should be optimized. While several examples of enzymatic reaction optimization have been described previously for colorimetric- [33,34,35] and SERS-based measurements [15] of HRP with the oPD substrate, no studies have focused on understanding and refining the SERS detection procedure. For the first time, the current paper addresses this issue. Using silver nanoparticles (AgNPs) and model mixtures of oPD and DAP, we successively and rationally evaluated the effects of the pH, the concentration of the aggregating agent, and the particle surface chloride stabilizer. As a result, we strongly narrowed the range of optimal detection conditions to a mildly acidic pH of 3–4 using a 0.93–1 M citrate buffer and AgNPs stabilized with 20 mM chloride. With these optimal conditions, we have demonstrated a two orders of magnitude advantage in the detection limits of SERS compared to colorimetry for both DAP and HRP. The paper is intentionally rich in appendices, with some of them providing additional information about the behavior of the studied system and the remainder being dedicated to the detailed description of the experiments for easier reproduction of the protocol.

## 2. Results

### 2.1. Synthesis, Characterization, and Standardization of Silver Nanoparticles

Silver nanoparticles are known to be slowly oxidized by atmospheric oxygen. In order to obtain repeatable results over the entire study, fresh batches of AgNP colloids were routinely synthesized using a well-established hydroxylamine method [36]. To minimize oxidation, they were stored in a reaction medium containing excess hydroxylamine and used within 36 h after the synthesis. UV–vis spectra were used for the rapid characterization of each batch. To standardize the particle size, samples with a maximum of plasmonic band between 406.6 and 408.6 nm (Figure 1a) were selected for further use. The number-weighted mean hydrodynamic diameter of the synthesized particles, measured using nanoparticle tracking analysis (NTA), was found to be 39 ± 1 nm (Figure 1b). The particle shape was characterized using the combination of two techniques: transmission electron microscopy (TEM) and atomic force microscopy (AFM), as they complement each other. While TEM provides adequate information about the shape of a particle’s projection along the XY plane, it lacks data on the Z direction. On the other hand, although the extraction of a particle’s lateral dimensions from AFM data is not straightforward due to convolution with the tip shape and minor XY drifts of the sample, the heights of rigid silver particles on a rigid silicon support are measured with sub-nm accuracy. For the TEM data (Figure 1c), each particle’s projection (N = 500) was measured in two directions: the longest axis, with a mean of 39.1 nm and standard deviation (SD) of 7.8 nm, and the shortest one, with a mean of 33.9 nm and SD of 6.1 nm. The average aspect ratio was 1.16. The average particle height determined from AFM (Figure 1d) was 35.1 nm with an SD of 8.7 nm (N = 122). For additional information about the particle shape, please refer to Appendix A.

As long as the reaction medium contains excess hydroxylamine, which is a reactive compound that could cause some side reactions, it is preferred to transfer AgNPs into a standard medium for the SERS measurements. During the synthesis, particles are stabilized with a chloride anion [36] provided by hydroxylamine hydrochloride, so the most natural way is to keep this stabilizer and remove all other components. This was performed using double centrifugation and pellet resuspension in a NaCl solution. In order to standardize the particle concentration, the AgNP colloid absorption at the plasmonic maximum was adjusted to 14.25 (0.95 at a 15-fold dilution). As evidenced by the NTA data (Figure 1b), this procedure causes only a minor shift in the mean particle size from 39 to 40 nm, which is expected due to the loss of the smallest particles to the supernatant. The total particle concentration also does not change within the NTA technique uncertainty. The samples of standardized AgNPs were used for the SERS measurements within 5 h to prevent particle oxidation due to atmospheric oxygen.

### 2.2. Estimation of HRP Effective K_M_ for oPD

Although we have claimed that our main goal is the optimization of SERS measurements of DAP, this procedure is dependent on the conditions of the preceding enzymatic reaction. SERS spectra with AgNPs were previously reported for both oPD [15,27,37,38] and DAP [15,27,31,32], implying that they both have some affinity for the silver surface, so during the SERS measurements, they compete for binding to the metal. Thus, our goal could be refined as a search for conditions that favor the binding of small quantities of enzymatically formed DAP over the excess of unreacted oPD. In order to estimate the highest concentration of oPD that could be passed to the SERS detection stage from the enzymatic stage, the effective Michaelis constant for oPD was measured colorimetrically at 421 nm in 100 mM sodium phosphate buffer (pH 6.0) and a H_2_O_2_ concentration of 80 μM, which has been previously reported to be optimal in terms of the background reaction rate [35]. The 95% confidence interval for K_M_ was found to be 37 ± 3 μM. From this value and also visually from the graph (Figure 2 and Appendix B), it could be concluded that an oPD concentration of 1 mM (around 27 × K_M_) is the reasonable maximum. Indeed, additional oPD causes an increase of only a few percent in the enzymatic product formation rate but interferes with DAP during a subsequent SERS detection step.

### 2.3. Selection of pH for SERS Detection of DAP

An HRP enzymatic reaction is usually performed at a pH of 5 to 7; however, sensitive SERS detection of the produced DAP in its mixtures with unreacted oPD may have a different optimal pH than from within this range. As long as DAP has a pronounced SERS spectrum over a broad range of pHs (see Appendix C), hypothetically, several strategies may be considered:The enzymatic reaction is stopped by an acid or base, and this mixture is further added to the AgNP colloid for SERS measurements. The stop reagent should create the optimal pH for DAP binding to the silver surface and the optimal ionic strength for the proper aggregation of AgNPs.If the sensitive detection of DAP in its mixtures with oPD is possible at the pH optimum of HRP, the enzymatic reaction mixture could be directly added to the AgNP colloid, followed by SERS detection. The advantage of this approach is the absence of dilution by the stop reagent.

In this section and related appendices, we extensively analyze which processes interfere with DAP detection at various pHs for the rational exclusion of unviable strategies.

The first factor is the presence of unreacted H_2_O_2_ in reaction mixtures, which could cause the partial dissolution of AgNPs [4,39,40]. Using spectrophotometric measurements of the position and absorbance in the plasmonic band maximum, it has been shown (see Appendix D) that at an alkaline pH of 9, a measurable dissolution of AgNPs occurs within 10 min of incubation with 100 μM H_2_O_2_. Thus, stopping the enzymatic reaction by shifting to an alkaline pH is an unfavorable strategy.

The second factor is the presence of unreacted oPD. A two-step process was used to estimate the feasibility of DAP detection in its mixtures with oPD at different pHs. During the first step, aimed at narrowing the range of prospective pHs, a screening was performed using default concentrations of buffers/media (0.32 M for pH 9.1 and 0.5 M for other pHs). The pH is considered prospective if it meets two criteria. First, the most intense DAP bands should be detectable at a moderate DAP concentration of 100 nM. Second, their intensity should increase at 500 nM DAP. The screening results (Figure 3) show that in the presence of 1 mM oPD, DAP could be detected at pHs from 4 down to a strongly acidic conditions. Notably, the sensitivity at this range is high enough to detect weak DAP bands in the absence of added DAP. They originate from the minor quantities of DAP present in stock oPD. At a higher pH, oPD dominates the SERS spectrum, with no detectable DAP bands. As long as 1 mM of oPD was determined to be the highest reasonable concentration to use in the enzymatic reaction, the same screening was also performed at a lower oPD concentration of 0.1 mM (Figure 4). The qualitative picture is the same, with the prospective range extended up to pH 7.1 due to the weaker competition of oPD with DAP for binding to the silver surface. Another important feature is that at pH ≥ 5, the oPD spectrum is concentration dependent and contains bands with poorly reproducible intensities, thus providing an unsteady background for DAP detection. All the features mentioned in the oPD spectra are discussed in detail in Appendix E.

At the second step of pH selection, potential candidates from the screening (pH from 0.3 to 4) were directly compared to each other. In order to use the best possible conditions for each pH, the concentrations of buffers/media serving as aggregating agents for the AgNPs were optimized (Appendix F). The results of this comparison are shown in Figure 5.

The highest SERS signal for both of the most intense DAPH^+^ bands (733 and 1374 cm^−1^) in the presence of 1 mM oPD was observed at a mildly acidic pH of 3. Hence, the general procedure for the SERS detection of HRP activity should include the enzymatic reaction, stopped by the addition of concentrated citrate buffer with pH = 3 up to the final concentration of 0.93 M for the optimal aggregation of AgNPs. This sample could be directly mixed with AgNPs in a 1:1 ratio, with SERS measurement 2 min later.

### 2.4. Influence of AgNP Chloride Stabilizer

The last factor to consider in the current paper is the impact of AgNP surface modification on the SERS detection of DAP. This topic is truly inexhaustible, so only one system was studied as an example of this approach. The chloride anion has a strong affinity for the silver surface and provides the negative charge of AgNPs. In all previous experiments, AgNPs stabilized with 5 mM NaCl were used. This concentration could be varied in a narrow range from 2.5 to 20 mM, with AgNPs being unstable outside this range on a timescale of several hours. Below 2.5 mM, there is not enough chloride on the surface to fully stabilize the colloid. Above 20 mM, destabilization occurs due to the contraction of the electrical double layer by the increased ionic strength.

The data provided in Figure 6 evidence that the SERS intensity for both DAP bands asymptotically increases with the saturation of the silver surface by chloride. The highest NaCl concentration of 20 mM, which still results in stable AgNPs, should be used in the assay.

### 2.5. Comparison of Colorimetric and SERS Detection Methods for DAP and HRP

In order to evaluate the performance of the optimized conditions for DAP measurement, colorimetric and SERS detection methods were directly compared to each other (Figure 7). Both assays were performed in a 0.93 M citrate buffer at pH 3. The oPD concentration was chosen to be 0.33 mM with the following considerations: The maximum concentration of the stock citrate buffer is around 1.5 M. At higher concentrations, it starts to precipitate if the room temperature drops below 20 °C. This obstacle strongly deteriorates the repeatability if thin and poorly visible white sediment is overlooked. Thus, to stop the enzymatic reaction, the sample should be diluted around three times with stock 1.5 M citrate, which turns 1 mM of oPD from the enzymatic reaction into 0.33 mM on a SERS detection stage. It should be noted that a pH of 3 is optimal not only for SERS but also for colorimetry, as the molar absorption coefficient of DAPH^+^ is slightly higher than that of DAP (21,400 vs. 17,000 M^−1^ × cm^−1^ [41]). Thus, the comparison of both techniques in these conditions is justified.

The calibration curve for colorimetric detection (Figure 7b) was linear, with a negative deviation at high DAP concentrations above 18 μM. The limit of detection (LOD) was estimated to be 478 nM. The mean coefficient of variation (CV) for the range above the LOD was 6%.

Both concentration curves for the SERS measurements of DAP (Figure 7c,d) were strongly nonlinear (S-shaped). Saturation at high concentrations is typical for SERS measurements due to the occupation of all available binding sites on the silver surface. A negative deviation from linearity at low concentrations is less expected. Two hypotheses could be proposed for this phenomenon. First, the issue might be kinetic: at low concentrations, two minutes is not enough to reach equilibrium for DAPH^+^ adsorption onto the silver surface. On the other hand, it could be an indication of a cooperative process, namely, the oligomerization of DAP on the surface, which has been shown for DAP on MoSe_2_ [42].

In order to describe these datasets with calibration curves, an appropriate function was found. It includes the exponential term to account for the concave shape at low concentrations, together with the hyperbolic term, characteristic of processes with saturation. The estimated LODs for the SERS detection of DAP were 2.2 and 4.9 nM for 733 and 1374 cm^−1^ bands, respectively. These values are around two orders of magnitude lower than the LOD of the colorimetric technique. The mean CVs for the range above the LOD were measured to be 24% and 22%, respectively.

Finally, the same comparison was made for HRP (Figure 8). As long as the optimization of the enzymatic reaction stage is beyond the scope of the current study, for this demonstration, some typical conditions were used: room temperature, pH 6, 1 mM oPD, 80 μM H_2_O_2_, and a reaction time of 10 min. The reaction was stopped using a three-fold dilution with citrate buffer (1.5 M at pH = 3). The chosen conditions influence both detection techniques (SERS and colorimetry) equally; hence, the estimate for the ratio of the two LODs is unbiased.

The calibration curve for the colorimetric HRP measurement (Figure 8a) was nonlinear, as reported previously [33]. Instead of limiting ourselves to a narrow linear range, an appropriate nonlinear function was found to describe all data points. The estimated LOD for HRP was 6.5 pM, and the mean CV for the range above the LOD was 11%. The calibration curves for the SERS detection of HRP have the same S-shape as those for DAP, with estimated LODs of 0.067 and 0.32 pM for 733 and 1374 cm^−1^ bands, respectively. Therefore, as with the DAP measurements, the SERS detection of HRP at 733 cm^−1^ is about two orders of magnitude more sensitive compared to colorimetry, which confirms the performance of the proposed SERS-based detection system. The mean CVs for the range above the LOD were measured to be 13% and 9% for 733 and 1374 cm^−1^ bands, respectively. Notably, the SERS signal repeatability in HRP measurements was substantially higher (and comparable to the colorimetry) than in the model DAP-oPD mixtures. This could be caused by the low grade of commercially available DAP (>90%) with the unavoidable impurity of 2-hydroxy-3-aminophenazine [43]. Retrospectively, we suppose that despite higher uncertainties, the model DAP-oPD mixtures were highly useful during optimization as long as key features (calibration curve shape and SERS to colorimetry LODs ratio) were confirmed for enzymatically generated DAP.

## 3. Discussion

The measurement of HRP activity is the basis of the catalytic enhancement and detection system in an ELISA and also in some other analytical protocols. Several published papers describe the usage of an oPD substrate together with the SERS detection of a DAP product [14,15,23,27,28]. The SERS-based detection of DAP in the mixture after an HRP-catalyzed enzymatic reaction strongly depends on the conditions, and the current study addresses this topic, which has generally been neglected previously.

The AgNP colloid used in the current study is typical for SERS applications. It has been synthesized using a well-established hydroxylamine method [36] with a slightly increased hydroxylamine hydrochloride concentration (2.4 instead of 1.5 mM). This excess decreases the oxidation of AgNPs during short-term storage and also improves the resuspension of the pellet after the centrifugation due to better surface stabilization with a higher chloride concentration. Freshly synthesized colloids were used during the entire study, leaving the problem of long-term AgNP stability to later research.

The particle shape is slightly prolonged with a mean aspect ratio of 1.16, which corresponds well to the value of 1.13 independently measured in our previous study [31]. About 8% of particles in total have a specific shape in TEM images: rods, triangles, squares, and a drop-like shape (Figure A4). The mean hydrodynamic spherical equivalent diameter (Figure 1b) is about 3–4 nm larger than the average microscopic size (Figure A2), which is a typical value for the thickness of two electrical double layers (one at each side of the particle) at an ionic strength of 5 mM. Therefore, most of the particles in the colloid are individual and not aggregated.

As long as fresh batches of AgNPs were used during the study, special attention was paid to their standardization. The mean particle size was standardized through the selection of synthesized colloids by the position of plasmonic band maximum in the UV–visible spectrum (Figure 1a). The medium and particle concentration of the colloid were standardized using double centrifugation and resuspension of the pellet in NaCl solution, followed by an adjustment of absorption at the plasmonic band maximum. The default NaCl concentration was chosen to be 5 mM and was later optimized to 20 mM (Figure 6). Using NTA, it was shown that double centrifugation and resuspension did not cause particle aggregation (Figure 1b). In order to preserve the desired nanoparticle concentration, the absence of their adsorption onto the laboratory plastic (Appendix G) was carefully monitored.

As mentioned previously, the SERS measurement of HRP consists of two stages: the enzymatic reaction and SERS detection of the resulting DAP. The optimization problem for this system is complicated, as these stages may require different conditions but are at the same time coupled, because the reaction mixture is passed from the first stage to the second. In order to develop a highly sensitive protocol, our attention was focused primarily on the second stage, because its performance changes hugely depending on the conditions, and the basic underlying principles were unknown. For the oPD–DAP system, both substances are commercially available. This occasion strongly simplifies the research, as it could be made with well-controlled model mixtures.

The concentration of oPD has a direct impact on the enzymatic reaction rate. At the same time, during the SERS detection, the unreacted oPD interferes with the measurement of small quantities of the produced DAP. With this contradiction in mind, first, the highest reasonable oPD concentration for the enzymatic reaction was estimated to be around 1 mM and used for further experiments.

The pH of SERS detection stage turned out to be the key factor affecting the sensitivity. Summarizing all the findings, DAP has a high affinity for the silver surface across the entire pH range from 0.3 up to 9.1 (Appendix C). At pH 9, AgNPs are prone to dissolution by unreacted H_2_O_2_ (Appendix D). The influence of oPD is complex. In neutral form (which exists at pH ≥ 3 according to its pK_a_ = 4.55 [44]), it exerts a moderate affinity for the silver surface (Appendix E). No other form (oPDH^+^ or oPDH_2_^2+^) was found in the SERS spectra. The neutral oPD competes with DAP for binding with silver, so at 1 mM oPD, even a relatively high DAP concentration of 500 nM is not detectable at pH > 5 (Figure 3). Second, at pH > 4, minor amounts of silver (I) present in the AgNPs facilitate the oxidation of oPD (Appendix E). The product of this process is not DAP, but most likely some intermediate on a route from oPD to DAP. The bands of this oxidation product (or products) are numerous in the SERS spectra and vary in their intensities, creating a poor background for DAP detection at pH > 4. Finally, the stock oPD contains minor quantities of DAP, both in a commercial powder and additionally formed due to spontaneous oxidation during oPD solution storage prior to use (Figure A18). This nonenzymatic DAP could deteriorate the sensitivity of the assay.

Taking into account all the listed obstacles, the sensitive SERS detection of DAP in the presence of 1 mM oPD is possible at pH ≤ 4. After the optimization of the aggregating agent concentration at each pH, a direct comparison showed the highest sensitivity at pH 3 (Figure 5). At higher pHs, DAP experiences competition with oPD for binding to the silver. At lower pHs, DAPH^+^ is partially converted into DAPH_2_^2+^ according to its pK_a_ of around 1 [41]. The coexistence of two forms of analyte typically deteriorates the SERS detection sensitivity, because the substance is split between two sets of bands. From this point of view, pH 3, being equidistant from both pK_a_s (~1 and 5.1), is a “sweet spot”, maximizing the fraction of a single form, in this case, DAPH^+^.

The optimized conditions for DAP detection rationally found in the current study are superior to those used by chance previously. In the pioneering work by Dou et al. [14], the mixture, after 20 min of enzymatic reaction at pH 5, was directly added to AgNPs. This study, although very important as the first demonstration of the principle, suffered from a poor sensitivity despite the advantage of the resonant regime with a 514.5 nm excitation. In two recent works by Fu et al. [15,27], the enzymatic reaction was stopped using sulfuric acid, and the resulting mixture was added to AgNPs. Our optimized conditions are estimated to be around 10 times more sensitive in terms of DAP detection: a five-fold difference between sulfuric acid and pH 3 (Figure 5) and an additional two-fold improvement due to modification of AgNPs with 20 mM chloride (Figure 6).

A cross-study comparison is typically not straightforward due to numerous differences in protocols. To evaluate the performance of the proposed detection system, it has been directly compared to the reference colorimetric technique. This was first performed for model mixtures of DAP with 1 mM oPD and then repeated with HRP-generated DAP. These experiments revealed an important feature of the SERS-based detection system: the calibration curve is strongly nonlinear. This is not uncommon for modern commercial assays and analytical devices with automatic signal processing. At the same time, further research would be favorable, aimed at improving the curve slope in a low concentration range. Despite these peculiarities, the proposed SERS-based detection system exhibits around two orders of magnitude lower LODs (733 cm^−1^ band) than traditional colorimetry.

The resulting LOD for HRP of 0.067 pmol/L (1.3 amol per assay) places the developed approach for SERS-based HRP measurements into a category of highly sensitive techniques, along with fluorimetry [1,2,45] and chemiluminescence [3,46,47,48]. There is obviously plenty of room for further improvements regarding the following: optimization of the enzymatic reaction stage (pH, time, temperature, buffer type, etc.), replacement of near-spherical AgNPs with anisotropic objects (rods [49], nanoplates [50], etc.), which were shown to have higher enhancement factors, and others. The proposed protocol is also compatible with classic HRP enhancement procedures such as polymeric peroxidase, tyramide signal amplification, and so on. Therefore, we suppose that this HRP–oPD system with SERS detection using AgNPs has great potential for applications in highly sensitive ELISAs.

As for the basic research, additional insights into the causes of high DAP affinity for the silver surface would be advantageous for better understanding the driving forces of the proposed analytical system. In our study, oPD provided some clues about its orientation on the silver surface thanks to the nontrivial pH dependence of SERS spectra and the presence of the broad and intense band around 983 cm^−1^ of the NH_2_ group, covalently bound to silver. On the contrary, the reasons for a high DAP affinity for silver are vague. It exhibits intense SERS spectra with AgNPs in all three forms: DAP, DAPH^+^, and DAPH_2_^2+^. Therefore, protonation does not interfere with its binding to the surface. Despite the fact that DAP and oPD both have two adjacent NH_2_ groups, the affinity of DAP for silver is much higher. See, for instance, Figure 4, for spectra at pH 6, where 100 nM DAP is visible in the presence of 100 μM oPD (1000 times the difference in concentration). The SERS spectrum of DAP also lacks the analog of the 983 cm^−1^ band of oPD for any of its forms. From these facts, our best guess is that DAP has some other orientation on the surface than oPD does. A probable faint hint is given by the concave shape of dependence in the low concentration region (Figure 7c,d), which may indicate a cooperative process of DAP oligomerization during adsorption. This hypothesis agrees with the existence of molecule ribbons in crystals of both DAP and DAPH^+^ [51,52,53], but it is hard to experimentally confirm or reject it for the surface of silver nanoparticles in a liquid. To conclude, if the proposed SERS-based HRP measurement system is to be widely used, this problem will require further research.

## 4. Materials and Methods

### 4.1. Reagents

NH_2_OH·HCl (# H9876, ≥99%), AgNO_3_ (# 209139, ≥99%), oPD (# P9029, ≥98%), DAP (# 661376, 90%), highly stabilized salt-free HRP (# P2088, RZ 2.6–3.4, 200–300 pyrogallol units/mg), NaCl (# S9625, ≥99%), bovine serum albumin (BSA, # A7030, ≥98%), trisodium citrate dihydrate (# S4641, ≥99%), anhydrous sodium acetate (# 1.06268, ≥99%), sodium dihydrogen phosphate (# 71496, ≥99%), sodium phosphate dibasic (# 1.06586, ≥99%), and (3-aminopropyl)-trimethoxysilane (# 281778, 97%) were purchased from Sigma-Aldrich. To minimize the oxidation of oPD, freshly purchased powder was stored at −20 °C instead of recommended +4 °C. Chemically pure NaOH (>99%) for AgNP synthesis and extra-pure grade H_2_SO_4_ were purchased from Chimmed, Russia. Chemically pure glacial acetic acid, extra-pure HCl (35–38 weight %), boric acid, and citric acid hydrate were purchased from Component-Reaktiv, Russia. Biotechnology-grade glycine (# Am-O167, >99%) was purchased from Helicon, Russia. Sodium tetraborate decahydrate (# 31457, 99.5%) was purchased from Fluka. All solutions were prepared using deionized water (18.2 MΩ·cm) from the MilliQ UF Plus system (Millipore, Molsheim, France).

### 4.2. Synthesis and Standardization of AgNP Colloids

AgNPs were synthesized using the hydroxylamine method by Leopold and Lendl [36], with a modified hydroxylamine hydrochloride concentration as previously described [31]. Briefly, 1 mL of 10 mM AgNO_3_ was quickly added to 9 mL of 2.56 mM hydroxylamine hydrochloride with 3.33 mM NaOH in a 15 mL polypropylene test tube (refer to Appendix G for notes) and immediately stirred using a vortex mixer. Typically, five to seven tubes were prepared at once for further selection. The mixture was stored for 1 h to complete the reaction.

Each colloid was characterized with a UV–vis absorbance spectrum (300–750 nm) in a 15-fold dilution with water using a Shimadzu UV-1800 spectrophotometer. To standardize the particle size, samples with a maximum of plasmonic band between 406.6 and 408.6 nm were selected for further use within 36 h. Prior to the SERS experiments, 1.5 mL aliquots of colloid were centrifuged for 10 min at 7000 RPM (Heraeus Sepatech Biofuge A). These conditions were chosen on this particular rotor for the highest AgNP yield as a compromise between sedimentation and resuspension completeness. The sediment was resuspended in 1.5 mL of 5 mM NaCl, followed by intensive vortex mixing. Next, the AgNP colloid was subjected to the second centrifugation, followed by resuspension in 0.75 mL of the desired NaCl concentration (2.5 to 20 mM). The absorbance at maximum (A_max_) was measured for this around a two-fold concentrate (using a 30-fold dilution in water) and adjusted by dilution in NaCl up to A_max_ = 14.25 (0.95 × 15) to standardize the particle concentration. Standardized AgNP colloids were used for SERS within 5 h to prevent their oxidation with atmospheric oxygen.

### 4.3. Determination of Mean Hydrodynamic Diameter and Total Particle Concentration of AgNPs

Some of the colloids were characterized with nanoparticle tracking analysis (NTA) using the Nanosight LM10 HS instrument (NanoSight, Amesbury, UK) in the following configuration: 405 nm, 65 mW laser unit with passive temperature readout, a high-sensitivity EMCCD-type camera, and NTA software version 2.3 build 33 software. The sample was diluted with 5 mM NaCl 15,000 or 13,000 times to reach the optimal concentration for the NTA technique. Twenty-one videos of particles’ Brownian motion (60 s each) were recorded using the following camera settings in advanced mode: shutter = 450, gain = 180, lower threshold = 455, and higher threshold = 16,380. All videos were processed in basic mode, with a detection threshold of 6 (Multi) and automatic settings for other options. The data from all repeats (at least 3600 tracks) were merged to obtain the particle size distribution, mean number-weighted hydrodynamic diameter, and total particle concentration, corrected for the dilution factor.

### 4.4. Transmission Electron Microscopy of AgNPs

One side of a nitrocellulose, carbon-coated PELCO^®^ Cu grid (Ted Pella Inc., Redding, CA, USA) was silanized by applying a 20 mkL droplet of a freshly prepared, 1% water solution of (3-aminopropyl)-trimethoxysilane (APTMS) for 2 min, followed by triple washing in water. The grid was placed into an as-synthesized AgNP colloid for 10 min for particle adsorption due to electrostatic forces between negatively charged particles and positively charged amino groups of the APTMS coating. After that, the grid was washed three times with water and dried. The control sample contained no nanoparticles, only surface silanization. Grids were examined at 80 kV with a JEM-1400 transmission electron microscope (JEOL, Tokyo, Japan) equipped with a Quemesa CCD camera (Olympus Soft Imaging Solutions, Münster, Germany). The images were manually processed in ImageJ 1.51n (National Institutes of Health, Bethesda, MD, USA) by measuring the largest and smallest dimensions for each individual particle.

### 4.5. Atomic Force Microscopy of AgNPs

A 5 × 5 mm piece of Si wafer was rinsed with water and immersed into a freshly prepared 1% water solution of APTMS for 2 min. The sample was transferred into water and vortexed to wash off the excess APTMS. This washing step was performed three times in total. The sample was purged with clean air and immersed in an as-synthesized AgNP colloid for 3 min for particle adsorption. After that, the sample was washed three times as described before and purged with clean air. The control sample contained only surface silanization and triple washing. The samples were examined with an Asylum MFP-3D-SA atomic force microscope (Asylum Research, Santa Barbara, CA, USA) in tapping mode on air, using silicon NSG03 cantilevers (NT-MDT, Moscow, Russia) with a tip curvature radius of 10 nm. The resonant frequency of the cantilever was 145.45 kHz. The force constant determined by the built-in passive thermal vibration method was 1.44 N/m. The amplitude set point was kept at 99.4–99.6% of free amplitude, and the tip velocity was set to 200 nm/s to avoid particle movement with the tip. After obtaining eight 1 × 1 μm topographic images, the absolute free amplitude of the cantilever was estimated to be 26.5 ± 1.0 nm (mean ± SD, N = 3) in the area with no particles using approach–retraction curves. In order to flatten the support level, the acquired topographic data were treated by the sequence of the following built-in procedures: (1) subtraction of the first-order plane, (2) subtraction of the first-order plane with particles masked, (3) subtraction of the mean from each line in the fast scan direction with particles masked, and (4) subtraction of the first-order polynomial from each line in the fast scan direction with particles masked. The heights of individual particles were measured using a central section method.

### 4.6. Colorimetric Estimation of HRP Effective K_M_ for oPD

The enzymatic reaction was carried out at room temperature with 80 μM of H_2_O_2_ in 100 mM sodium phosphate buffer (pH 6.0) with 5 μg/mL BSA to prevent HRP loss. The formation of DAP was followed in the kinetic regimen by measuring the absorbance at 421 nm using a UV-1800 (Shimadzu, Kyoto, Japan) spectrophotometer. After mixing the buffer concentrate, H_2_O, H_2_O_2_, and oPD in a single-use, 1 cm acrylic cuvette, the reaction was started by the addition of an HRP aliquot, followed by pipetting for 30 s and measurements of absorbance for 350 s, one point per second. During the preliminary experiments, an appropriate concentration of HRP was chosen (8 ng/mL) so that the absorbance after 5 min at 5 mM oPD did not exceed 0.25, being in a linear range for DAP colorimetric detection. Background oPD oxidation was measured for each oPD concentration the same way as the enzymatic reaction, but with the HRP aliquot replaced by buffer. The resulting pairs of kinetic curves (with HRP and without) were processed to obtain the enzymatic reaction rate (see Appendix B for details). The DAP concentration from A_421_ was calculated using an absorption coefficient of 15,560 M^−1^×cm^−1^ at pH 6. The entire dataset of the reaction rate versus the oPD concentration was fitted using the Michaelis–Menten equation in Mathematica 10.2 (Wolfram Research, Champaign, IL, USA) using the built-in NonlinearModelFit function with the weight equal to inversed data variance at each oPD concentration.

### 4.7. Preparation of Standard DAP Solutions

DAP is hardly soluble in water (70 to 100 μM depending on the temperature), so its exact solution cannot be prepared by dissolving the weighted sample. Instead, an excess amount (around 20–30 mg) of DAP was added to 12 mL of water, stirred on a vortex mixer, and sonicated for 30 min in a 40 W ultrasonic bath to create a saturated solution. After cooling to room temperature, the suspension was filtered through a 0.22 μm PES syringe filter, with the first 1 mL being withdrawn. The DAP concentration was measured colorimetrically at a five-fold dilution in water (to be in a linear range) at 421 nm, using an absorption coefficient of 15,560 M^−1^×cm^−1^. The resulting solution was stored in a dark test tube due to the light sensitivity of DAP and used the same day.

### 4.8. Procedure for SERS Measurements

As the developed analytical procedure is proposed for broad usage, all the data provided in the paper were collected using the portable spectrometer i-Raman Pro BWS475–785 H (BWTek, Plainsboro, NJ, USA) with a 785 nm excitation and 20× objective. For validation purposes during the qualitative comparison of Raman or SERS spectra, the second spectrometer, N’tegra Spectra (NT-MDT, Moscow, Russia) with a 785 nm excitation and 100× objective, was used.

An aliquot (20 μL) of sample was added to 20 μL of AgNPs (with the aggregation timer being started) and mixed by pipetting. A droplet of the mixture (20 μL) was placed on the surface of thick Al foil. The focus of the spectrometer was positioned at the top surface of the droplet and then shifted 600 μm into the liquid. Two minutes after aggregation had started, the SERS spectrum was acquired. To avoid detector saturation on samples with strongly different SERS intensities, the spectrometer setup was chosen to be as follows: 0.5 s collection time with 30 automatically averaged repeats (15 s total collection time) at 135 mW of power on the sample.

Buffers and media for different pHs were prepared from the following components: H_2_SO_4_ was diluted from concentrated acid; the pH 2 buffer was prepared from glycine and H_2_SO_4_ instead of the more common HCl to avoid the influence of chloride on the SERS signal; the pH 3 and 4 buffers were prepared from citric acid and trisodium citrate; the pH 5 buffer was prepared from acetic acid and sodium acetate; the pH 6 and 7 buffers were prepared from monosodium and disodium phosphates; and the pH 9 buffer was prepared from boric acid and sodium tetraborate.

### 4.9. Colorimetric and SERS Measurements of DAP at pH 3

In order to minimize the amount of DAP formed in the oPD solution due to spontaneous oxidation, only freshly dissolved oPD stock was used. Samples containing DAP in 1 M citrate buffer (pH 3) with 0.33 mM oPD were placed into a 96-well polystyrene plate (Greiner, # 655080) in four technical replicates (200 μL), mixed for 10 s for proper meniscus settling, and measured at 454 nm with an xMark plate reader (Bio-Rad, Tokyo, Japan). Alternatively, similar samples with lower DAP concentrations were used for the SERS measurements. Each curve was independently repeated with another oPD and DAP stocks, polystyrene plate, or AgNP batch to obtain a realistic estimate of the data variance.

### 4.10. Colorimetric and SERS Measurements of HRP

The substrate mixtures (1 mM oPD and 80 μM H_2_O_2_ in 100 mM citrate buffer at pH 6, with 5 μg/mL of BSA to prevent HRP losses at low concentrations) were prepared in test tubes with the same precaution for the oPD stock as for the DAP measurement. The reaction was initiated by the addition of HRP aliquots. After ten minutes, the reaction was stopped by the addition of twice the volume of citrate buffer 1.5 M (pH 3) and either placed into a 96-well polystyrene plate in four technical replicates (200 μL), followed by measurement with the xMark plate reader at 454 nm, or used for SERS. Each curve was independently repeated with another oPD stock, HRP dilutions, polystyrene plate, or AgNP batch to obtain a realistic estimate of the data variance.

### 4.11. Processing of Concentration Curves

The data of all concentration curves (DAP and HRP) were fitted in Mathematica 10.2 using the built-in LinearModelFit or NonlinearModelFit functions with the weight equal to inversed data variance at each DAP or HRP concentration. The limit of detection (LOD) was calculated from the calibration curve as the concentration, corresponding to the signal equal to the mean of the blank + 3 standard deviations of the blank. As long as the standard deviation strongly fluctuates for small sample sizes (N), the mean coefficient of variation (CV) for the range above the LOD was used as an estimator of assay repeatability. For each concentration, the signal SD was calculated and normalized by the difference of the mean signal and mean blank. The resulting CVs were averaged for all concentrations above the LOD.

### 4.12. Processing of Raman and SERS Spectra

For presentation purposes, the polynomial background was subtracted from each spectrum using OPUS 7.0 software (Bruker Optik GmbH, Ettlingen, Germany) through the “Baseline correction” built-in function.

To extract quantitative information about band intensities from the SERS spectra, two procedures were used in the current study (see Appendix H for additional figures).

The first one contains background correction only. It has been applied to spectra obtained under varying conditions (aggregating agent concentration optimization, pH selection, and surface chloride optimization). First, two pieces were cut from each spectrum: 670–805 cm^−1^ for the 733 cm^−1^ band and 1174–1740 cm^−1^ for the 1374 cm^−1^ band (Figure A25a). Next, for each piece, two ranges for background estimation to the left and right from the desired band were chosen: 683–700 and 794–805 cm^−1^ for 733 cm^−1^ and 1174–1214 and 1695–1720 cm^−1^ for 1374 cm^−1^ (Figure A25b,c). A linear fit of the background was made for these ranges and subtracted from the cut pieces (Figure A25d,e). Finally, the intensities of the desired bands were taken at single pixels, corresponding to 733.28 and 1373.65 cm^−1^. For the curves obtained in H_2_SO_4_, the maximum intensity between 729.95 and 734.95 cm^−1^ was used, because the position of the band experienced some shifts due to the varying ratio of DAPH^+^ to DAPH_2_^2+^.

The second procedure is more rigorous and was applied to the spectra obtained in optimized conditions only (DAP and HRP calibration curves). In this case, all the bands in the spectra have the same shape but vary in their intensities. Thus, an approach similar to the quantitative Rietveld analysis of XRD diffractograms could be exploited to reduce the noise and extract weak signals. In a preliminary stage, using several spectra at a high DAP concentration (high signal-to-noise ratio), the desired and a few neighboring bands were fitted with either a general Pearson IV peak shape or a Gaussian for weak peaks. After that, these peak shapes were fixed and normalized to have unity intensity at their maximums (see equations in Appendix H). The non-negative linear superposition of these found peak shapes, together with the quadratic polynomial background, could be used to describe any spectrum obtained in optimized conditions. In the main stage, the same pieces of spectra as for the first procedure were fit with this superposition using the built-in NonlinearModelFit function in Mathematica 10.2 (Figure A26).

### 4.13. Statistical Analysis

All calculations of statistical parameters were performed using the built-in functions of the Mathematica 10.2 package. The confidence intervals for the mean values in replicate NTA measurements were estimated using a Student t-distribution (MeanCI function) with a confidence level of 95%. A Mann–Whitney test (MannWhitneyTest function) was used for sample comparison. The differences were considered significant at *p* < 0.05.

## 5. Conclusions

The current paper is the first to describe the optimization of the SERS detection stage for the HRP–oPD–DAP system using hydroxylamine silver colloids. The pH has been shown to be the key parameter defining the sensitivity of the SERS detection of DAP in the presence of excess oPD. At a pH > 3, DAP experiences competition with oPD for binding to the silver. At a pH < 3, DAPH^+^ is partially converted into DAPH_2_^2+^, which also deteriorates the sensitivity. At an optimal mildly acidic pH of 3, a 0.93–1 M citrate buffer, and AgNPs stabilized with 20 mM chloride, an advantage of two orders of magnitude in the LODs for SERS compared to colorimetry were demonstrated for both DAP and HRP. The resulting LOD for an HRP of 0.067 pmol/L (1.3 amol per assay) makes this detection system a useful tool for the development of highly sensitive, SERS-based ELISA protocols.

## Figures and Tables

**Figure 1 molecules-29-00793-f001:**
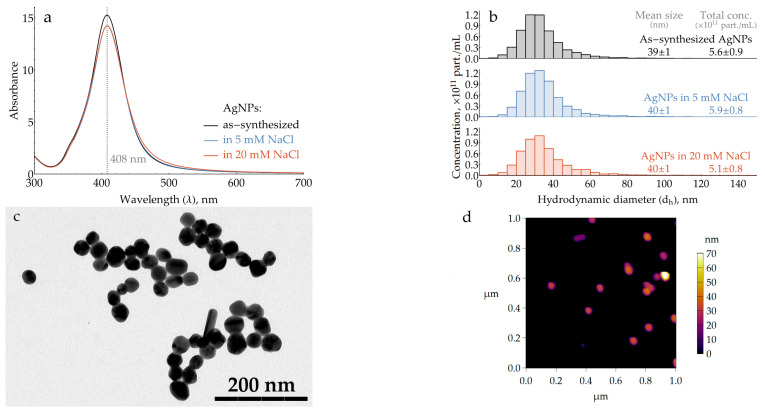
Characterization of AgNP colloids. (**a**) Typical UV–visible absorption spectra. (**b**) Hydrodynamic diameter and particle concentration measured with nanoparticle tracking analysis. Data for panels (**a**) and (**b**) are provided for both “as-synthesized” particles and those transferred into 5 or 20 mM NaCl. Numerical data in panel (**b**) are presented as 95% confidence intervals. (**c**) Transmission electron micrograph of AgNPs. (**d**) Atomic force topography of AgNPs on silanized silicon wafer.

**Figure 2 molecules-29-00793-f002:**
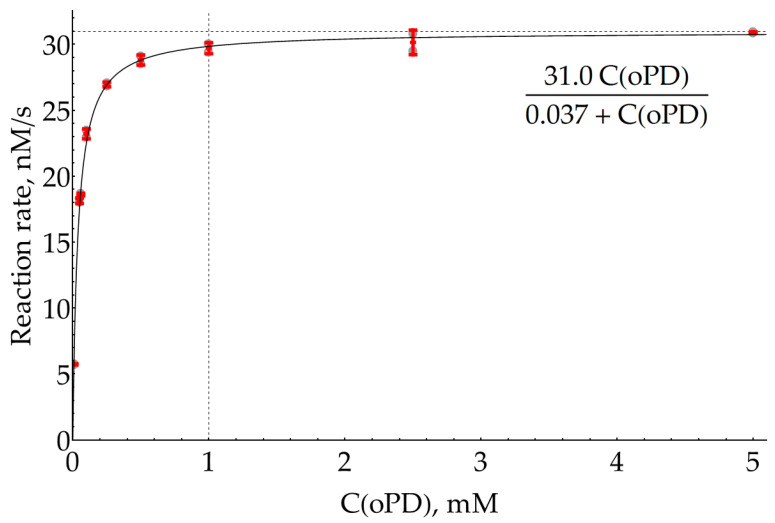
Dependence of HRP enzymatic reaction rate on oPD concentration measured colorimetrically at 421 nm in 100 mM sodium phosphate buffer (pH 6.0) with 8 ng/mL HRP and 80 μM H_2_O_2_. Open gray circles represent raw data (N = 2); red bars denote the mean ± standard deviation. A solid line represents the best fit.

**Figure 3 molecules-29-00793-f003:**
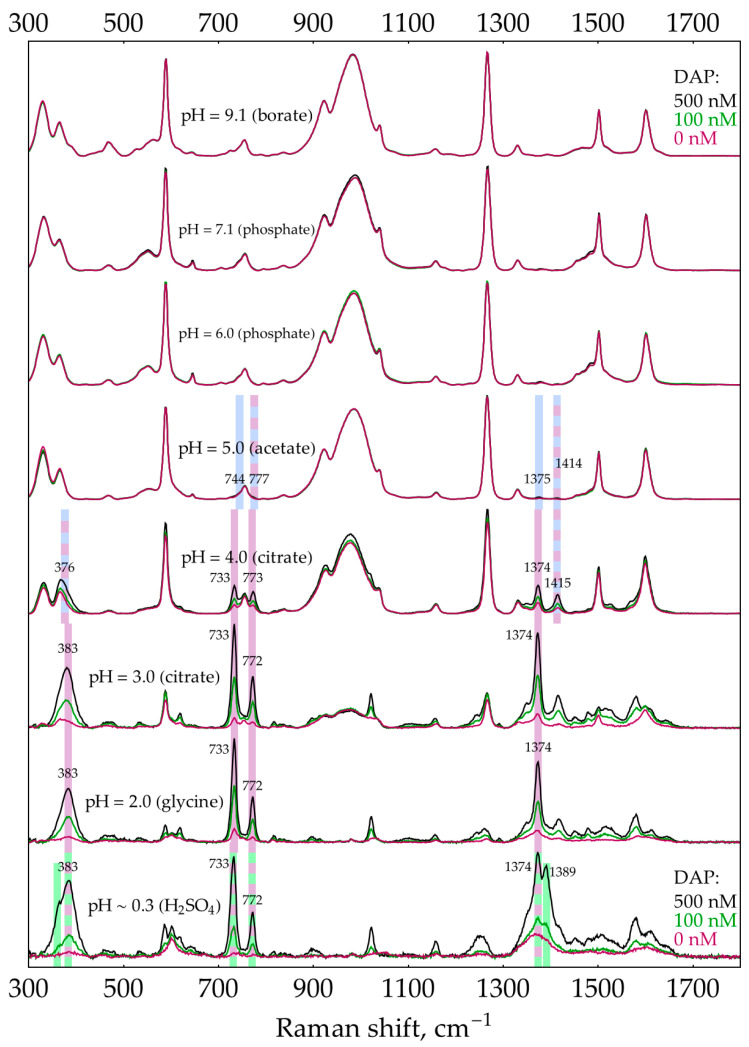
SERS spectra of 0, 100, and 500 nM DAP in the presence of 1 mM oPD with AgNPs in 5 mM NaCl at different pHs. The polynomial baseline was subtracted from each spectrum. All three spectra at each pH were normalized to the most intense band of all DAP concentrations at this pH. The shading and numbers represent the most intense bands of different forms of DAP: blue—DAP, purple—DAPH^+^, and green—DAPH_2_^2+^. For the reference spectra of oPD and DAP at different pHs, please refer to Figure A8 and Figure A15, respectively.

**Figure 4 molecules-29-00793-f004:**
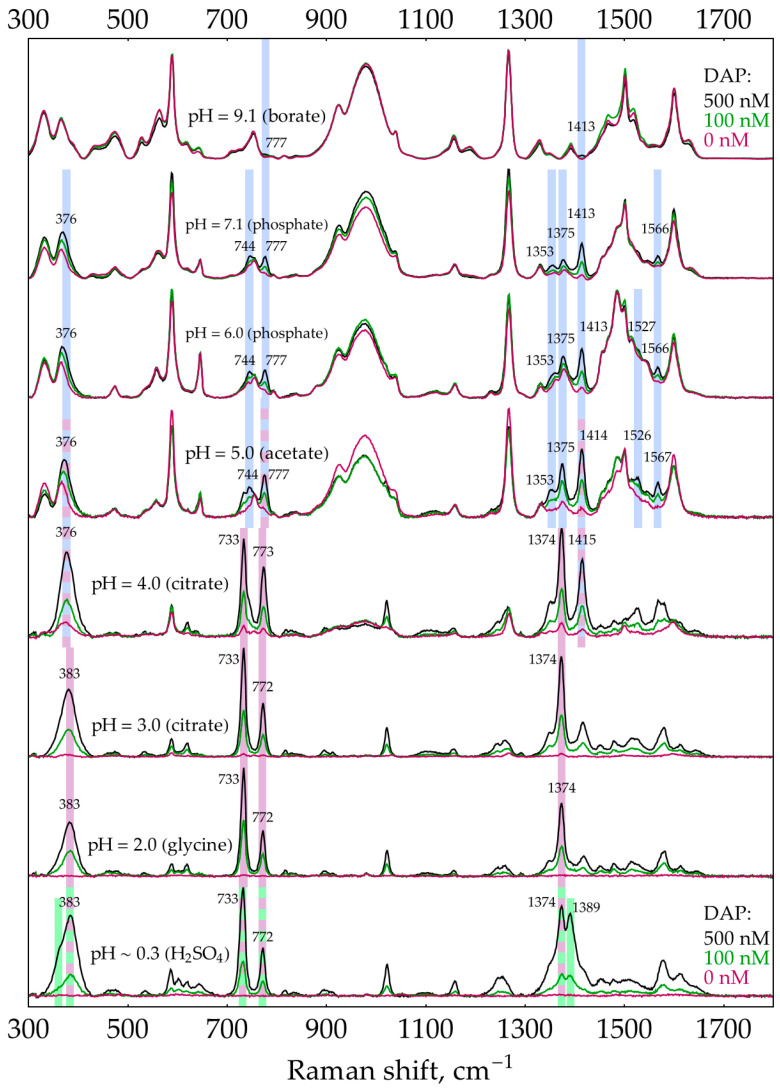
SERS spectra of 0, 100, and 500 nM DAP in the presence of 0.1 mM oPD with AgNPs in 5 mM NaCl at different pHs. The polynomial baseline was subtracted from each spectrum. All three spectra at each pH were normalized to the most intense band of all DAP concentrations at this pH. The shading and numbers represent the most intense bands of different forms of DAP: blue—DAP, purple—DAPH^+^, and green—DAPH_2_^2+^. For the reference spectra of oPD and DAP at different pHs, please refer to Figure A8 and Figure A16, respectively.

**Figure 5 molecules-29-00793-f005:**
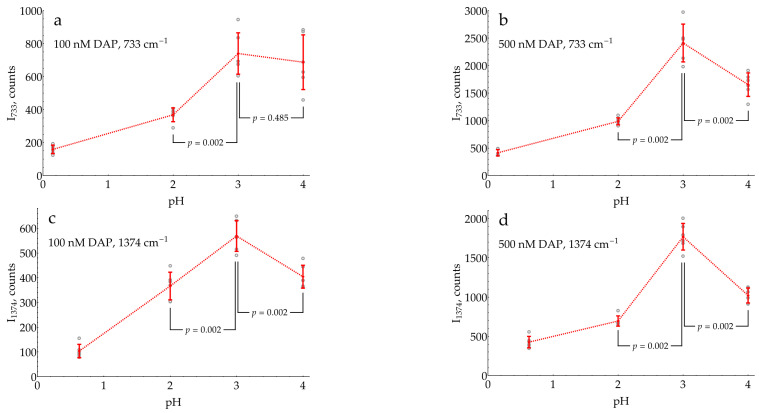
Comparison of SERS intensities of DAP bands in DAP mixtures with 1 mM oPD at different pHs. The concentrations of buffers/media used for AgNP aggregation are listed in Appendix F. For each data point, a signal from a corresponding blank sample without added DAP was subtracted. (**a**,**b**) SERS intensities at 733 cm^−1^ with 100 or 500 nM DAP, correspondingly; (**c**,**d**) SERS intensities at 1374 cm^−1^ with 100 or 500 nM DAP, correspondingly. Open gray circles represent raw data (N = 6); red bars denote mean ± standard deviation. Reported *p*-values result from the Mann–Whitney test. Dashed lines connect the means for visual clarity.

**Figure 6 molecules-29-00793-f006:**
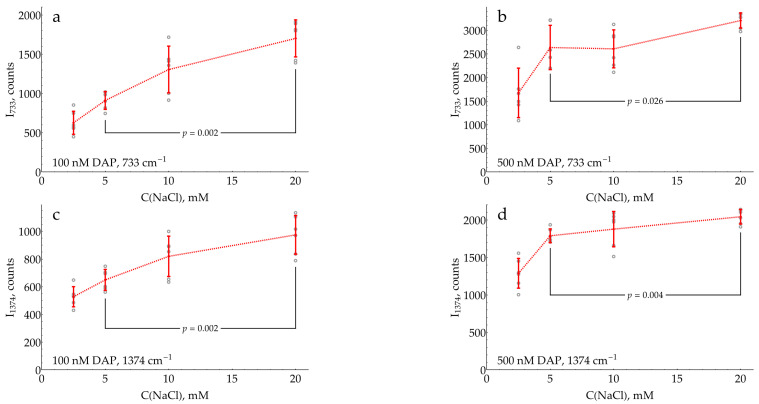
SERS intensities of DAP bands in DAP mixtures with 1 mM oPD, measured with AgNPs stabilized by varying NaCl concentrations and aggregated with 0.93 M of citrate buffer at pH 3. For each data point, a signal from a corresponding blank sample without added DAP was subtracted. (**a**,**b**) SERS intensities at 733 cm^−1^ with 100 or 500 nM DAP, respectively; (**c**,**d**) SERS intensities at 1374 cm^−1^ with 100 or 500 nM DAP, respectively. Open gray circles represent raw data (N = 6); red bars denote mean ± standard deviation. Reported *p*-values result from the Mann–Whitney test. Dashed lines connect the means for visual clarity.

**Figure 7 molecules-29-00793-f007:**
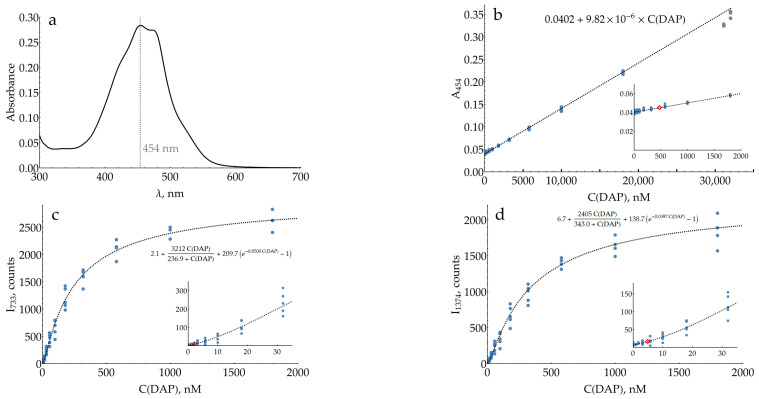
Measured absorption and SERS intensities for different DAP concentrations in DAP mixtures with 0.33 mM oPD in 0.93 M citrate buffer at pH 3. (**a**) Absorption spectrum of 18 μM DAP; (**b**) absorption at 454 nm measured using a plate reader with sample volumes of 200 μL (N = 29 for blanks and 8 for other concentrations); (**c**) SERS intensity at 733 cm^−1^ (N = 20 for blanks and 5 for other concentrations); (**d**) SERS intensity at 1374 cm^−1^ (N = 20 for blanks and 5 for other concentrations). Insets represent the regions of low concentrations, with the LOD marked with a red diamond. Dashed lines represent best fits.

**Figure 8 molecules-29-00793-f008:**
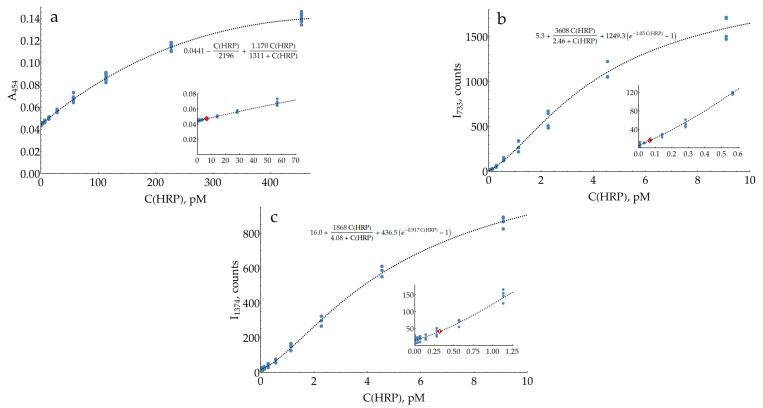
Absorbance and SERS intensities for different HRP concentrations. The enzymatic reaction was performed for 10 min at pH 6, 1 mM oPD, and 80 μM H_2_O_2_. It was stopped using three-fold dilution with 1.5 M citrate buffer at pH 3. (**a**) Colorimetric detection at 454 nm measured using a plate reader with total sample volumes of 200 μL (N = 12 for blanks and 8 for other concentrations); (**b**) SERS intensity at 733 cm^−1^ (N = 19 for blanks and 4 for other concentrations); (**c**) SERS intensity at 1374 cm^−1^ (N = 19 for blanks and 4 for other concentrations). Insets represent the regions of low concentrations, with the LOD marked with a red diamond. Dashed lines represent best fits.

## Data Availability

All the relevant data are provided in the main text or appendices.

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
