# Peer review of "Highly Sensitive Measurement of Horseradish Peroxidase Using Surface-Enhanced Raman Scattering of 2,3-Diaminophenazine"

_molecules, 2024, doi:10.3390/molecules29040793_

Round 1

Reviewer 1 Report

Comments and Suggestions for Authors

1. how about the SERS test consistency?

2. it is better to add more material charazations.

Author Response

We are thankful to reviewer for detailed analysis of the original manuscript. Valuable comments and questions helped us to identify missing pieces of information and substantially improve the paper. Below we provide a response to every comment or question as well as a description of the corresponding changes made in the manuscript.

  1. How about the SERS test consistency?

In order to characterize the repeatability of the proposed SERS-based method for HRP measurement, the mean coefficient of variation (CV) for the range above the LOD was used. CV for each concentration was calculated as the ratio of SD to pure signal (the difference of the mean signal and mean blank). The resulted CVs were averaged for all concentrations above LOD as described in Section 4.11. This measure was calculated to be 13% and 9% for 733 and 1374 cm-1 DAP bands, respectively, which was comparable to 11% for colorimetric determination of HRP.

  1. It is better to add more material charazations.

Silver nanoparticle colloids used in the present study were synthesized by a well-established hydroxylamine method (reference [36] in revised manuscript), widely used for SERS applications. In the original manuscript, nanoparticles were characterized by UV-visible absorption, hydrodynamic particle size, and particle concentration. In addition, in the revised manuscript a particle size and shape were extensively evaluated using a combination of two techniques: transmission electron microscopy and atomic force microscopy.

A brief list of key changes:

  • Title, abstract and introduction were substantially rewritten. Graphical abstract was updated to include the LOD for HRP in terms of quantity per assay.
  • The Conclusion section was added to the manuscript.
  • Several technical mistakes were fixed
    • Captions to Figures 7 and 8: during experiments, more repeats were done for blanks compared to other concentrations for proper estimation of blank SD. Captions were updated to include this information.
    • Information about storage temperature for oPD was added to the Section 4.1
    • Shift in the focus position of the microscope in Section 4.8 was changed from 300 to 600 μm, which is a correct value.
  • Additional characterization of AgNPs was performed de novo using transmission electron microscopy and atomic force microscopy. Section 2.1, Figure 1, and Discussion were updated. Two new sections were added into Materials and Methods to describe these experiments (4.4 and 4.5). Appendix A was added for additional information about the particle size and shape.
  • Section 4.11 in Materials and methods was added to describe the calibration curve processing, LOD and CV calculations.
  • All graphs in the main part of the manuscript together with corresponding graphs in the appendices were redrawn according to recommendations from the reviewers (thicker lines, better data readability, error bars, numbers for band positions, etc.)
  • English language was improved throughout the manuscript.

Reviewer 2 Report

Comments and Suggestions for Authors

The entitled manuscript “Improved detection of 2,3-diaminophenazine for horseradish peroxidase quantification using surface-enhanced Raman spectroscopy” by Evgeniy G. Evtushenko et al. quantified 2,3-diaminophenazine for horseradish peroxidase using ELISA coupled SERS. I regret that the result of the paper because there is no improvement of the detection, just changed pH condition. I do not recommend publishing this paper. 

Author Response

We are thankful to the reviewer for detailed analysis of the original manuscript. Valuable comments and questions helped us to identify missing pieces of information and substantially improve the paper. Below we provide a response to every comment or question as well as a description of the corresponding changes made in the manuscript.

The entitled manuscript “Improved detection of 2,3-diaminophenazine for horseradish peroxidase quantification using surface-enhanced Raman spectroscopy” by Evgeniy G. Evtushenko et al. quantified 2,3-diaminophenazine for horseradish peroxidase using ELISA coupled SERS. I regret that the result of the paper because there is no improvement of the detection, just changed pH condition. I do not recommend publishing this paper.

The current study is devoted to measurement of HRP using a SERS detection of its enzymatic product, 2,3-diaminophenazine (DAP) with AgNPs. More specifically, the study focuses on the performance of SERS detection stage, which has not been studied previously, to our best knowledge. In general, there are several papers dedicated to this system. The pioneering work on this topic was published in 1997 by Dou, et al (reference [14] in revised manuscript). Although the product of this enzymatic reaction was wrongly assigned to azoaniline (nowadays it has been solely identified as DAP [29,30]), it was the first principal demonstration of this approach. In this paper, the solution after the enzymatic reaction was mixed with AgNP colloid in 1:9 ratio followed by SERS measurement in 5 seconds. No optimization or justification of used conditions was provided by the authors. Next three papers by Hawi, Nithipatikom, and coauthors [24-26] in 1998-2002 utilized this approach for immunohistochemistry. HRP-labeled antibodies with oPD substrate and SERS detection were used to quantify various antigens in cell cultures. Silver colloids were added into the wells with an enzymatic reaction mixture directly or after the pre-aggregation of AgNPs with 1 M NaCl. Except this pre-aggregation, there were no attempts to optimize the SERS detection conditions. The next paper in 2006 [23] was devoted to glucose measurements but utilizes the product of HRP-oPD reaction with a SERS detection. Again, the product was wrongly assigned to azoaniline. The system contained oPD, glucose and gold nanoparticles with glucose oxidase and HRP adsorbed on their surface all mixed together followed by SERS measurement. Reaction time and temperature were optimized for this one-pot system. These two parameters influence the enzymatic reactions rather than SERS efficiency. Finally, in 2021 and 2022 two papers were published by Fu and coauthors [27,15]. The first one was devoted to the measurement of H2O2 based on HRP-oPD reaction with SERS detection. The second one describes the ELISA using the same detection system with slight modifications. In both papers the enzymatic reaction was stopped by sulfuric acid followed by addition of AgNPs and SERS detection. The second paper contains the optimization of the enzymatic reaction (H2O2 concentration, time, pH and temperature), but no data were provided about the optimization of the SERS detection stage. To conclude, although HRP-oPD reaction with SERS detection was successfully used to create different analytical protocols (including several examples of ELISAs), and this sequence of papers demonstrates the steady progress of this approach, especially two latest papers by Fu and coauthors, optimization and deep understanding of conditions for SERS detection stage may boost this system even further. The current study is intended to do so.

The effects of pH, aggregating agent concentration and AgNP surface chloride stabilizer were extensively evaluated. Indeed, pH was found to be a key factor affecting the sensitivity of DAP measurement in the presence of excess unreacted o-phenylenediamine (oPD) substrate. We have shown that at pH > 4 the sensitivity is strongly deteriorated by the competition between DAP and oPD for binding to the silver surface. SERS spectra were also complicated by multiple bands of variable intensity which originated from partial oxidation of oPD by minor amounts of Ag(I) present in AgNPs (see Appendix E for details). At pH < 3, the sensitivity decreased as well due to partial transformation of protonated DAP into doubly-protonated form. Therefore, the current study not only contains the optimized conditions (in other words, a protocol for sensitive SERS measurements of HRP) but also provides chemical background for this analytical system which makes it a highly useful tool for further research in this area. Another valuable piece of information in our study is the accurately determined LOD for HRP (0.067 pmol/L or 1.3 amol per assay). To our best knowledge, it is the first demonstration that SERS measurement of HRP is comparable in sensitivity to the best examples of such highly sensitive detection systems as enhanced chemiluminescence [3,46-48] and fluorimetry [1,2,45], which is highly promising for the development of SERS-based ELISAs.

Does the introduction provide sufficient background and include all relevant references? – Must be improved

The introduction has been substantially changed. One reference was removed and 13 were added into this section.

Are all the cited references relevant to the research? – Can be improved

References were carefully evaluated and updated.

Is the research design appropriate? - Must be improved

Additional AgNP characterization was added to improve the research design.

Are the methods adequately described? - Must be improved

Section 4.11 in Materials and methods was added to describe the calibration curve processing, LOD and CV calculations.

Are the results clearly presented? - Can be improved

All graphs in the main part of the manuscript together with corresponding graphs in the appendices were redrawn according to recommendations from the reviewers (thicker lines, better data readability, error bars, numbers for band positions, etc.).

Are the conclusions supported by the results? - Must be improved

The Conclusion section was added to the manuscript.

A brief list of key changes:

  • Title, abstract and introduction were substantially rewritten. Graphical abstract was updated to include the LOD for HRP in terms of quantity per assay.
  • The Conclusion section was added to the manuscript.
  • Several technical mistakes were fixed
    • Captions to Figures 7 and 8: during experiments, more repeats were done for blanks compared to other concentrations for proper estimation of blank SD. Captions were updated to include this information.
    • Information about storage temperature for oPD was added to the Section 4.1
    • Shift in the focus position of the microscope in Section 4.8 was changed from 300 to 600 μm, which is a correct value.
  • Additional characterization of AgNPs was performed de novo using transmission electron microscopy and atomic force microscopy. Section 2.1, Figure 1, and Discussion were updated. Two new sections were added into Materials and Methods to describe these experiments (4.4 and 4.5). Appendix A was added for additional information about the particle size and shape.
  • Section 4.11 in Materials and methods was added to describe the calibration curve processing, LOD and CV calculations.
  • All graphs in the main part of the manuscript together with corresponding graphs in the appendices were redrawn according to recommendations from the reviewers (thicker lines, better data readability, error bars, numbers for band positions, etc.)
  • English language was improved throughout the manuscript.

Reviewer 3 Report

Comments and Suggestions for Authors

The manuscript is not suitable for publication in its current format.

1. The title and abstract do not accurately represent the content of the manuscript. They are vague and confusing.

2. The nanoparticles need to be characterized using additional techniques such as XRD, TGA, and...

3. Instead of adding all repetitions in the graphs, the average and the standard deviation must be used for drawing the graphs.

4. The method must be validated using real samples.

Comments on the Quality of English Language

Extensive editing of English language required.

Author Response

We are thankful to the reviewer for the critical analysis of the original manuscript. Valuable comments and questions helped us to identify missing pieces of information and substantially improve the paper. Below we provide a response to every comment or question as well as a description of the corresponding changes made in the manuscript.

  1. The title and abstract do not accurately represent the content of the manuscript. They are vague and confusing.

The title and abstract were improved.

  1. The nanoparticles need to be characterized using additional techniques such as XRD, TGA, and...

The current study employs a well-established hydroxylamine synthesis of AgNPs (reference [36] in revised manuscript), which is widely used for SERS applications. According to the original protocol, slight variations in reagent mixing speed during silver nitrate addition could result in batch-to-batch differences in particle size and particle concentration. Potentially, particle shape could also experience some variation. In our original manuscript, we characterized particle size and concentration using two methods. UV-visible absorption is fast but indirect. The position of absorption maximum is related to mean particle size, while absorption in this maximum is related to particle concentration. The second method was nanoparticle tracking analysis, which directly reported hydrodynamic particle size and total particle concentration. For the revised manuscript, we’ve additionally evaluated particle size and shape by a combination of two techniques: transmission electron microscopy and atomic force microscopy. The results of this newly performed TEM characterization on shape (average aspect ratio of 1.16) were in good agreement with our previous paper [31] (average aspect ratio of 1.13) made a year ago. It means that the chosen procedure of AgNP synthesis followed by selection based on position of absorption maximum consistently results in fresh batches of very similar particles.

As for proposed XRD and TGA (thermogravimetric analysis), both of these techniques require large sample quantities, at least for instruments available for us: 300-500 mg for XRD and 50 mg for TGA. These quantities are almost impossible to collect taking into account the batch volume of 10 mL and silver concentration of 1 mM (0.1 mg/mL). We can’t increase the batch volume as it will definitely change the rate of reagent mixing, resulting in different nanoparticle size distribution and particle concentration.

  1. Instead of adding all repetitions in the graphs, the average and the standard deviation must be used for drawing the graphs.

We agree that after data acquisition, all repetitions should be statistically analyzed to estimate appropriate measures of central tendency (e.g., mean) and dispersion (e.g., standard deviation). But as for graphical data presentation, a trend of the last 6-7 years is to encourage reporting of individual data points. One of the key arguments for this is that mean and SD hide the peculiarities of data distribution: asymmetry, heavy tails, potential bimodality as well as signs of non-independence (non-zero covariance). As a result, some top-ranked journals, for instance, Nature updated their editorial policy (https://www.nature.com/documents/nr-editorial-policy-checklist-flat.pdf). Data presentation section states that “Individual data points are shown when possible, and always for n ≤ 10”. MDPI has no recommendations on this topic, so we are following the best practices we know. As a result, Figures 2, 5a-d, 6a-d, A6, A7, A20-A23 were redrawn with individual data points kept as gray circles, and error bars (mean±SD) added in red. These error bars obscure the raw data, but it could be analyzed using full-size versions of the graphs. At the same time, for analytically relevant graphs (Figures 7b-d, 8a-c), we would like to keep individual data points with nothing obscuring them, so the data is available for visual analysis. Data repeatability for each of these graphs is present in text as a mean coefficient of variation (CV) for the range over the LOD.

  1. The method must be validated using real samples.

The key goal of the current paper was to optimize the SERS detection procedure for 2,3-diaminophenazine (DAP), catalytically generated from o-phenylenediamine by HPR because this specific problem has not been studied previously. Although this method already demonstrates high sensitivity with the LOD for HRP of 0.067 pmol/L or 1.3 amol per assay, it might be improved even further as it has been stated in the Discussion section. To obtain the best results in ELISA, we would like to continue the optimization in our next paper.

As for principal applicability of this method to real ELISA, in the current paper we could rely on existing studies. Hawi, Nithipatikom, and coauthors [24-26] have shown that HRP-labeled antibodies with oPD substrate and AgNPs-assisted SERS detection at pH = 5 could be used for quantification of various antigens in adherent cell cultures in a microplate. Fu and coauthors [15] described the microplate ELISA for human chorionic gonadotropin in both classical “sandwich” format and after the tyramide signal amplification step. SERS-based readout of enzymatically generated DAP was used with AgNPs at pH ≈ 0 (few moles/L of H2SO4). Conditions proposed in the current paper are in between these two cases (pH = 3, 0.93-1 M citrate buffer). The key differences include the way to stop the enzymatic reaction and the stabilizer of AgNPs.

(x) Extensive editing of English language required

Thank you for this comment. English language was improved using MDPI English editing services.

Does the introduction provide sufficient background and include all relevant references? – Must be improved

The introduction has been substantially changed. One reference was removed and 13 were added into this section.

Are all the cited references relevant to the research? – Must be improved

References were carefully evaluated and updated.

Is the research design appropriate? - Must be improved

Additional AgNP characterization was added to improve the research design.

Are the methods adequately described? - Must be improved

Section 4.11 in Materials and methods was added to describe the calibration curve processing, LOD and CV calculations.

Are the results clearly presented? - Must be improved

All graphs in the main part of the manuscript together with corresponding graphs in the appendices were redrawn according to recommendations from the reviewers (thicker lines, better data readability, error bars, numbers for band positions, etc.).

Are the conclusions supported by the results? - Must be improved

The Conclusion section was added to the manuscript.

A brief list of key changes:

  • Title, abstract and introduction were substantially rewritten. Graphical abstract was updated to include the LOD for HRP in terms of quantity per assay.
  • The Conclusion section was added to the manuscript.
  • Several technical mistakes were fixed
    • Captions to Figures 7 and 8: during experiments, more repeats were done for blanks compared to other concentrations for proper estimation of blank SD. Captions were updated to include this information.
    • Information about storage temperature for oPD was added to the Section 4.1
    • Shift in the focus position of the microscope in Section 4.8 was changed from 300 to 600 μm, which is a correct value.
  • Additional characterization of AgNPs was performed de novo using transmission electron microscopy and atomic force microscopy. Section 2.1, Figure 1, and Discussion were updated. Two new sections were added into Materials and Methods to describe these experiments (4.4 and 4.5). Appendix A was added for additional information about the particle size and shape.
  • Section 4.11 in Materials and methods was added to describe the calibration curve processing, LOD and CV calculations.
  • All graphs in the main part of the manuscript together with corresponding graphs in the appendices were redrawn according to recommendations from the reviewers (thicker lines, better data readability, error bars, numbers for band positions, etc.)
  • English language was improved throughout the manuscript.

Reviewer 4 Report

Comments and Suggestions for Authors

- there are many abbreviations in the paper; the reader sometimes has problems in remembering, was DAP, oPD, ... is. please add a abbreviation table of the most important abbreviations

- Typo in line 394: "standartization"....

- the sequence of chapters appears unusual: you have: "4. Materials and Methods" after introduction, resulst and discussion.

Would you please think about, to change this perhaps to the "normal" sequence.

Author Response

We are thankful to the reviewer for detailed analysis of the original manuscript. Valuable comments and questions helped us to identify missing pieces of information and substantially improve the paper. Below we provide a response to every comment or question as well as a description of the corresponding changes made in the manuscript.

There are many abbreviations in the paper; the reader sometimes has problems in remembering, was DAP, oPD, ... is. please add a abbreviation table of the most important abbreviations

Thank you for this recommendation. We have added the short abbreviation table at the beginning of the paper. However, we are not sure that this is allowed by the journal. So, it is up to the technical editor to keep this table or remove it entirely.

Typo in line 394: "standartization"....

Thank you for mentioning this typo. Is has been fixed.

The sequence of chapters appears unusual: you have: "4. Materials and Methods" after introduction, results and discussion. Would you please think about, to change this perhaps to the "normal" sequence.

This section order is defined by the journal template. Unfortunately, we can't do much about it.

A brief list of key changes:

  • Title, abstract and introduction were substantially rewritten. Graphical abstract was updated to include the LOD for HRP in terms of quantity per assay.
  • The Conclusion section was added to the manuscript.
  • Several technical mistakes were fixed
    • Captions to Figures 7 and 8: during experiments, more repeats were done for blanks compared to other concentrations for proper estimation of blank SD. Captions were updated to include this information.
    • Information about storage temperature for oPD was added to the Section 4.1
    • Shift in the focus position of the microscope in Section 4.8 was changed from 300 to 600 μm, which is a correct value.
  • Additional characterization of AgNPs was performed de novo using transmission electron microscopy and atomic force microscopy. Section 2.1, Figure 1, and Discussion were updated. Two new sections were added into Materials and Methods to describe these experiments (4.4 and 4.5). Appendix A was added for additional information about the particle size and shape.
  • Section 4.11 in Materials and methods was added to describe the calibration curve processing, LOD and CV calculations.
  • All graphs in the main part of the manuscript together with corresponding graphs in the appendices were redrawn according to recommendations from the reviewers (thicker lines, better data readability, error bars, numbers for band positions, etc.)
  • English language was improved throughout the manuscript.

Reviewer 5 Report

Comments and Suggestions for Authors

In this manuscript, the authors presented a deep study for the optimization of SERS conditions for the SERS detection of DAP and HRP, enzymatic product and enzyme label, respectively, used in ELISA assays. The article is very specific and the audience have to be expert in this topic. The authors have followed a clear and careful methodology which has been explained in detail in the article, mi congratulations for this way of working. Thus, I recommend, after minor revision, it can be considered for the final publication. Please see my detailed comments below:

1.     Abstract: the authors should explain the meaning of HRP.

2.     The authors should include updates reference (and not only self-references) to justify ¨the so-called SERS-based ELISA is growing scientific field which gradually builds up a range of available approaches as well as the number of successful applications¨.

3.     Many times (>10) appears the term SER and the correct word is SERS, please correct these typo mistakes.

4.     Line 81: the authors described that AgNP were transferred to NaCl. They should explain in this moment why they transfer the nanoparticles to this media and advantages or drawbacks.

5.     Authors should show more characterization of the AgNP: TEM images (as well as referred in the discussion section) or SEM images. Additionally, the authors should detail the shape of the nanoparticle.

6.     The color distinction of Figure 1b is too low to see the specific corresponding data, please change it.

7.     Figure 2: the authors should indicate the number of measurements performed per point to calculate/describe the curve and including the standard deviation per point.

8.     Figure 4: authors should include the band assignment. I was able to find part of them is in the appendix, nevertheless, I think this band assignment should appear in the main manuscript or the appendix should be references in a better way. In addition, the authors should include the reference spectra of oPD and DAP. My recommendation is also changed the color of the shadows, it is not easy to recognize each color.

9.      Figure 5, 6, 7, 8, the figure lines should be thicker.

10.  Line 223: authors should explain how the calculate the LOD and explain the parameter ¨coefficient variant (CV)¨.

11.  Materials and methods, line 401, authors should explain the term ¨sol¨ used many times a long the manuscript.

12.  Authors should include a Table comparing their LOD with the LODs published in previous scientific reports using SERS, colorimetry assays and a combination of both.

13.  Finally, I noticed section of ¨conclusions¨ is missing. In this section authors should summarize all the achievements reached through this article.

Author Response

We are thankful to the reviewer for thorough analysis of the original manuscript. Valuable comments and questions helped us to identify missing pieces of information and substantially improve the paper. Below we provide a response to every comment or question as well as a description of the corresponding changes made in the manuscript.

  1. Abstract: the authors should explain the meaning of HRP.

Thank you for this recommendation. We have added this abbreviation.

  1. The authors should include updates reference (and not only self-references) to justify ¨the so-called SERS-based ELISA is growing scientific field which gradually builds up a range of available approaches as well as the number of successful applications¨.

The original manuscript contained two references ([21,22] in the revised manuscript) to reviews to justify this statement. Larmour and coauthors (2010) review was devoted to a broader topic of enzyme measurements using SERS. The second section of our recent review (2023) was dedicated exclusively to SERS-based ELISA. To briefly summarize, five general approaches have been demonstrated. The first principal scheme is a direct one, similar to the current study, when the product of HRP or alkaline phosphatase reaction has more intense SERS spectrum than the substrate [7-9,11,12,14-19]. The second approach is ‘reversed’ and based on the measurement of decrease in enzymatic substrate concentration due to enzymatic reaction [6,20]. The third scheme was developed for the case when the enzymatic product is not SERS-active. The metal nanoparticles are modified to chemically capture this product and convert it into SERS-active substance [10]. The fourth scheme uses the tyramide HRP reaction to introduce dye-labeled metallic nanoparticles (so called “SERS-tags”) into vicinity of enzyme molecules [13]. Finally, the fifth approach is based on growth or dissolution of SERS-active nanoparticles by the substrate or the product of the enzymatic reaction [4,5], as long as SERS intensity depends on the size of metal nanoparticles.

The reference to our review was used deliberately to avoid the copying of the same references from our review to the current manuscript. Following the advice from the reviewer, all 17 relevant references [4-20] to original studies with SERS-based ELISAs were added to the text.

  1. Many times (>10) appears the term SER and the correct word is SERS, please correct these typo mistakes.

This abbreviation was used exclusively if the next word is spectrum/spectra (e.g., oPD SER spectrum) because SERS was defined as surface-enhanced Raman spectroscopy. Therefore, the expanded version for SERS spectrum was “surface-enhanced Raman spectroscopy spectrum” which is redundant. SER abbreviation is widely used. Google Scholar search results in around 2700 papers: https://scholar.google.com/scholar?q=%22SER+spectra%22|%22SER+spectrum%22

However, if this abbreviation is perceived as a typo for the advanced reader, we have redefined SERS as “surface-enhanced Raman scattering” which allowed us to change SER to SERS throughout the entire manuscript.

  1. Line 81: the authors described that AgNP were transferred to NaCl. They should explain in this moment why they transfer the nanoparticles to this media and advantages or drawbacks.

After the synthesis, the mixture contains AgNPs with the surface stabilized by chloride and the medium containing sodium nitrate, sodium chloride, possibly some products of hydroxylamine oxidation (other than N2) and excess hydroxylamine. The latter is useful for short-term storage (up to 36 hours) of nanoparticles for at least partial prevention of their oxidation by air oxygen. However, this excess hydroxylamine should be removed for SERS applications because it is a reactive compound. It has been shown that in strongly acidic conditions it reacts with DAP (see Figure A19). The most natural way is to remove all the components of the medium but retain the existing stabilizer, i.e., NaCl. A reasonable concentration of 5 mM was used by default. Later (see Figure 6) it was changed to 20 mM due to higher SERS intensities at this stabilizer concentration. This explanation was added to the manuscript.

  1. Authors should show more characterization of the AgNP: TEM images (as well as referred in the discussion section) or SEM images. Additionally, the authors should detail the shape of the nanoparticle.

We do not support the practice of republishing data from previous papers, so instead of providing the same TEM images that have been used in [31] to calculate the mean aspect ratio (referred in the Discussion section), we have done new TEM experiments from scratch. Additionally, we have characterized the particles with atomic force microscopy (AFM) as these two techniques complement each other. While TEM provides adequate information about the shape of a particle projection in the XY plane, it lacks the data on the Z direction. On the other hand, although extraction of particle lateral dimensions from AFM data is not straightforward due to convolution with tip shape and minor XY drifts of the sample, the height of rigid silver particles on a rigid silicon support is measured with sub-nm accuracy. The resulting images, average size characteristics, histograms and detailed analysis of the shape were added to section 2.1 and Appendix A.

The results of this newly performed TEM characterization on shape (average aspect ratio of 1.16) are in good agreement with our previous paper (average aspect ratio of 1.13) made a year ago. It means that the chosen procedure of AgNP synthesis followed by selection based on position of absorption maximum consistently results in fresh batches of very similar particles.

  1. The color distinction of Figure 1b is too low to see the specific corresponding data, please change it.

Thank you for this comment. This image was tough in terms of data presentation. All three histograms are very close to each other, and this is exactly what we wanted to show with this figure. Therefore, whichever colors are chosen, they do not allow clear visualization of every curve. As a result, we’ve redrawn this figure as a column of three histograms. We suppose that this version presents the data in a much more readable way.

  1. Figure 2: the authors should indicate the number of measurements performed per point to calculate/describe the curve and including the standard deviation per point.

The figure was redrawn to include mean and standard deviation. The information about the number of repetitions has been added to the caption. The same was done for Figures A6 and A7.

  1. Figure 4: authors should include the band assignment. I was able to find part of them is in the appendix, nevertheless, I think this band assignment should appear in the main manuscript or the appendix should be references in a better way. In addition, the authors should include the reference spectra of oPD and DAP. My recommendation is also changed the color of the shadows, it is not easy to recognize each color.

Thank you, this is a very valuable comment. We added relevant numeric values for Raman shifts into Figures 3, 4, A8, A11, A14-A17. As for adding reference spectra of DAP and oPD into Figures 3 and 4, we’re afraid that it’s not possible. At least 3 spectra should be added for DAP (neutral, protonated forms, and pH ≈ 0 containing a mixture of protonated and doubly-protonated forms) and 4-5 spectra for oPD as they are different at each pH (4, 5, 6, 7, and 9). This number of spectra definitely does not fit into a single page. As a solution, we’ve added a reference to Figures A8, A15, and A16 (DAP, 1 mM oPD and 0.1 mM oPD) into the captions for Figures 3 and 4 to simplify the navigation for the reader. Shadowing for all figures (3, 4, A8, A14, A15, A16, A17, and A19) were changed to more saturated colors.

  1. Figure 5, 6, 7, 8, the figure lines should be thicker.

The figures were redrawn. All lines were rendered thicker for better visibility.

  1. Line 223: authors should explain how the calculate the LOD and explain the parameter ¨coefficient variant (CV)¨.

The procedures used to calculate LOD and CV were included into section 4.11 of the Materials and Methods. The coefficient of variation (CV) is typically defined as signal SD/mean signal, therefore, representing a relative deviation from the mean. In order to account for a huge difference between blanks in colorimetry and SERS, we calculated this parameter in a slightly different and more correct way as SD/pure signal = SD/(mean signal – mean blank) as long as mean blank for colorimetry is a variable value depending on the quality of a polystyrene microplate and automatic data post-processing performed by the instrument.

  1. Materials and methods, line 401, authors should explain the term ¨sol¨ used many times along the manuscript.

The term sol is widely used in colloid and material science as a synonym to term ‘colloid’: https://scholar.google.com/scholar?q="nanoparticle+sol". To the best of our knowledge, it has been introduced by Thomas Graham in his 1894 paper on silicic acid solutions and colloids (https://doi.org/10.1039/JS8641700318). As this term raises questions, we have changed it to the more commonly used term ‘colloid’ throughout the entire manuscript.

  1. Authors should include a Table comparing their LOD with the LODs published in previous scientific reports using SERS, colorimetry assays and a combination of both.

To the best of our knowledge, there are no previously published LODs for SERS measurements of HRP. Although HRP-catalyzed reactions was used for SERS-based ELISA [6,14-20] as well as several other analytical protocols [23-28], there were no attempts to estimate the sensitivity of the detection system itself. The only paper, providing at least some dependence of SERS signal versus HRP concentration, is [18]. This paper is mainly dedicated to ELISA with SERS detection using TMB substrate. Figure 1B of this paper contains six SERS spectra with different HRP concentrations: 0, 0.02, 0.05, 0.1, 0.2, and 0.5 ng/mL. LOD was not calculated for this data, but we could roughly estimate it. Assuming a molar weight of 44 kDa for HRP, these concentrations correspond to 0, 0.45, 1.14, 2.27, 4.55, and 11.4 pmol/L. The spectrum for the blank contains the same bands as other spectra. The intensity for the lowest HRP concentration (0.45 pM) visually looks a bit higher than the intensity of the blank; therefore, we could use this value as a rough estimate. Real LOD could be even higher if the blank has a high SD. Anyway, this value is by order of magnitude higher than the LOD for HRP obtained in the current study (0.067 pM).

As for the colorimetric HRP detection, much more information is available. A total of 7 papers were found during an extensive literature search and summarized in the table below in chronological order.

Substrate

LOD and time

Authors and year

Reference

ABTS

0.25 pM, 50 min

Gallati, 1979

10.1515/cclm.1979.17.1.1

oPD

0.045 pM, 30 min

Gallati and Brodbeck, 1982

[34]

oPD

0.4*pM, 30 min

Bovaird, Ngo, and Lenhoff, 1982

[33]

MBTH + PDEA

2* pM, 15 min

Ngo, 1985

10.1007/BF02798440

TMB

0.25 pM, 30 min

Gallati and Pracht, 1985

10.1515/cclm.1985.23.8.453

oPD

0.87 pM, 20 min

Mekler and Bystryak, 1992

[1]

oPD

5* pM, 30 min

Fornera, et al., 2010

[35]

* lower limit of the linear range

Prior to comparison of these results to our LOD (6.5 pM), an important note should be made. All these results cannot be directly applied to real ELISA as they were obtained in a cuvette spectrophotometer with a large sample volume (at least 1 mL) and a very well-defined horizontal optical path of 1 cm. On the contrary, nowadays in the vast majority of cases ELISA is made in a 96-well microplate format with the vertical optical path of around 0.5 cm and, more importantly, with much higher signal variation. First, variations in sample volume caused by pipette directly affect the optical path length. Next, the absorbance of the well bottom varies from well to well. Finally, additional signal variation is caused by differences in meniscus settling. In the current paper, HRP was measured in the same way as real ELISA: 96-well microplate and 200 mkL of reaction mixture. From our experience, a factor of 4-10 should be applied to the LOD from the cuvette spectrophotometer to match the LOD from the plate reader.

The second notice should be made about the study by Gallati and Brodbeck (1982). It is a part of an original series of papers on chromogenic HRP substrates. Based on the results by Gallati and coauthors from 1979-1985, for almost a decade oPD was considered to be the most sensitive substrate for colorimetric HRP measurements (ABTS < TMB << oPD). Unfortunately, this superior sensitivity was not reproduced nether in study by Bovaird et al (1982), nor in the later study by Mekler and Bystryak (1990). Nowadays, all three mentioned substrates are widely recognized to be roughly of the same order of sensitivity. All of them are widely used in ELISA or antibody titer measurements with TMB prevalence due to slightly better sensitivity and, more importantly, due to the development of commercially available highly stabilized version which could be stored up to a year, while ABTS and oPD should be freshly prepared prior to use.

To conclude, for the microplate format the typical colorimetric sensitivity for HRP is in a picomolar/L range with a good correspondence to our result of 6.5 pM. We suppose that the addition of this table into the manuscript will definitely require a discussion of differences between spectrophotometer types, which is not quite relevant to the topic of the paper, namely, the SERS-based HRP measurement. Most likely, this table and discussion will confuse the reader rather than provide a tool for comparison.

  1. Finally, I noticed section of ¨conclusions¨ is missing. In this section authors should summarize all the achievements reached through this article.

We’ve added the Conclusions section which summarizes key findings of the study.

Does the introduction provide sufficient background and include all relevant references? – Can be improved

The introduction has been substantially changed. One reference was removed and 13 were added into this section.

Are the results clearly presented? - Can be improved

All graphs in the main part of the manuscript together with corresponding graphs in the appendices were redrawn according to recommendations from the reviewers (thicker lines, better data readability, error bars, numbers for band positions, etc.).

A brief list of key changes:

  • Title, abstract and introduction were substantially rewritten. Graphical abstract was updated to include the LOD for HRP in terms of quantity per assay.
  • The Conclusion section was added to the manuscript.
  • Several technical mistakes were fixed
    • Captions to Figures 7 and 8: during experiments, more repeats were done for blanks compared to other concentrations for proper estimation of blank SD. Captions were updated to include this information.
    • Information about storage temperature for oPD was added to the Section 4.1
    • Shift in the focus position of the microscope in Section 4.8 was changed from 300 to 600 μm, which is a correct value.
  • Additional characterization of AgNPs was performed de novo using transmission electron microscopy and atomic force microscopy. Section 2.1, Figure 1, and Discussion were updated. Two new sections were added into Materials and Methods to describe these experiments (4.4 and 4.5). Appendix A was added for additional information about the particle size and shape.
  • Section 4.11 in Materials and methods was added to describe the calibration curve processing, LOD and CV calculations.
  • All graphs in the main part of the manuscript together with corresponding graphs in the appendices were redrawn according to recommendations from the reviewers (thicker lines, better data readability, error bars, numbers for band positions, etc.)
  • English language was improved throughout the manuscript.

Round 2

Reviewer 2 Report

Comments and Suggestions for Authors

The paper has revised. I recommend publishing the journal.

Reviewer 3 Report

Comments and Suggestions for Authors

N/A